# Influence of Intraoperative Blood Loss on Tumor Recurrence after Surgical Resection in Hepatocellular Carcinoma

**DOI:** 10.3390/jpm13071115

**Published:** 2023-07-10

**Authors:** Suk-Won Suh, Seung Eun Lee, Yoo Shin Choi

**Affiliations:** Department of Surgery, College of Medicine, Chung-Ang University, Seoul 156-755, Republic of Korea; bumboy1@cau.ac.kr (S.-W.S.); selee508@cau.ac.kr (S.E.L.)

**Keywords:** hepatocellular carcinoma, intraoperative blood loss, blood transfusion, tumor recurrence, surgical resection

## Abstract

The high incidence of hepatocellular carcinoma (HCC) recurrence after surgical resection worsens the long-term prognosis. Besides tumor-related factors, operative factors such as perioperative blood transfusion have been reported to be related to HCC recurrence. However, excessive intraoperative blood loss (IBL) always necessitates blood transfusion, where IBL and blood transfusion may influence oncologic outcomes. We enrolled 142 patients with newly diagnosed single HCC who underwent hepatic resection between March 2010 and July 2021. Patients were stratified into two groups by IBL volume: Group A (IBL ≥ 700 mL, n = 47) and Group B (IBL < 700 mL, n = 95). The clinic–pathologic findings, operative outcomes, and cumulative probability of tumor recurrence and overall survival were compared between the two groups. In the study, increased IBL (1351 ± 698 vs. 354 ± 166, *p* < 0.001) and blood transfusion (63.8% vs. 6.3%, *p* < 0.001) were common in Group A, with a greater HCC recurrence (*p* = 0.001) and poor overall survival (*p* = 0.017) compared to those in Group B. Preoperative albumin (hazard ratio [HR], 0.471; 95% confidence interval [CI], 0.244–0.907, *p* = 0.024), microvascular invasion (HR, 2.616; 95% CI, 1.298–5.273; *p* = 0.007), and IBL ≥ 700 mL (HR, 2.325; 95% CI, 1.202–4.497; *p* = 0.012) were significant risk factors for tumor recurrence after surgical resection for HCC. In conclusion, efforts to minimize IBL during hepatic resection are important for improving long-term prognosis in HCC patients.

## 1. Introduction

Hepatocellular carcinoma (HCC) is a highly prevalent malignancy and common cause of cancer-related death worldwide [1]. Surgical resection is the only curative treatment among various loco-regional therapies for HCC that have shown favorable survival outcomes [2]. Widespread surveillance for detecting HCC in high-risk populations, comprising cirrhotic patients with hepatitis B virus (HBV) or hepatitis C virus (HCV) infection, has increased the rate of detection of early-stage HCC that is amenable to surgical resection [3]. However, the high incidence of HCC recurrence after surgical resection ensures an unsatisfactory long-term prognosis [4,5]. Clinico-pathologic factors related to the tumor, such as the tumor size and number, presence of microvascular invasion, poor histologic grade, and preoperative levels of α-fetoprotein (AFP) and prothrombin induced by vitamin K absence-II (PIVKA-II), have been reported to be associated with tumor recurrence after hepatic resection [6,7,8].

Despite recent improvements in surgical techniques, perioperative management, and dissection devices for hepatic resection, intraoperative blood loss (IBL) still remains a concern in patients who undergo hepatic resection [9]. The different hepatic vascular inflow and outflow systems confer challenges in controlling IBL from the hepatic vein during hepatic parenchymal transection and can lead to massive IBL that necessitates blood transfusion [10], which can adversely affect long-term oncologic outcomes of HCC patients [11,12]. However, few studies have been conducted to compare those based on the extent of IBL. Blood transfusion is always required for excessive IBL, and therefore the associated factors may be closely related, with a considerable overlap of the adverse effects. Based on a dose–response relationship between increased IBL and decreased disease-free and overall survival, a recent study identified increased IBL as a significant risk factor for HCC recurrence [13].

The aim of this study was to investigate the influence of operative factors, such as IBL and perioperative blood transfusion, on long-term oncologic outcomes, including tumor recurrence and overall survival, after surgical resection in HCC patients.

## 2. Patients and Methods

### 2.1. Patients

We enrolled 142 participants with well-preserved liver function who underwent curative surgical resection of newly diagnosed single HCC between March 2010 and July 2021 at our hospital. Patients with advanced-stage HCC, history of HCC treatment, borderline liver function, or insufficient clinical data were excluded from the analysis. Based on the IBL cutoff during hepatic resection, the participants were assigned to two groups: Group A (IBL ≥ 700 mL, n = 47) and Group B (IBL < 700 mL, n = 95). The primary outcome of this study was to compare the cumulative probability of tumor recurrence and overall survival between the two groups, and the secondary outcome was to identify risk factors after adjusting for several parameters that could potentially influence tumor recurrence and overall survival after surgical resection for HCC.

This study was approved by the Institutional Review Board of our institution (IRB No. 2023-006-19335) and was exempt from the requirement to obtain informed consent because accrual patient records were analyzed and no patient identification data were used.

### 2.2. Data Collection

Demographics, including sex, age, HBV or HCV infection, and body mass index (BMI), as well as preoperative laboratory results, including total bilirubin (TB), international normalized ratio (INR), and albumin, which represent liver function, were collected. Tumor characteristics, including tumor size and number, Edmonson and Steiner histologic grade, and the presence of microvascular invasion, as well as levels of tumor markers, such as AFP and PIVKA-II, were recorded. Operative data, including operative duration, proportion of major resection, IBL, requirement of blood transfusion, ICU admission rate, and hospital stay, were obtained. The type of hepatic resection was classified as minor resection or major resection, defined as resection of three or more segments (e.g., right hemi-hepatectomy, extended right hemi-hepatectomy, left hemi-hepatectomy, extended left hemi-hepatectomy, and central hepatectomy). Acute kidney injury (AKI) was defined in accordance with the 2012 Kidney Disease Improving Global Outcomes guidelines [14], which had higher predictability than other criteria for assessing prognosis: increase in serum creatinine by ≥0.3 mg/dL within 48 h; increase in serum creatinine to ≥1.5 times baseline within 7 days before surgery; or urine volume < 0.5 mL/kg/h for 6 h. Postoperative liver insufficiency was defined as a peak postoperative TB level of >7 mg/dL and/or the presence of ascites > 500 mL/day based on a previous study [15].

All patients were followed at 1, 3, and 6 months, and every 3 to 6 months thereafter as necessary. At every visit, imaging studies, such as computed tomography (CT), magnetic resonance imaging (MRI), or ultrasonography, and serologic tests, including tumor-marker analyses and biochemical tests for liver function, were performed. Follow-up was recorded as the period from surgery to the last visit, tumor recurrence, or death.

### 2.3. Anesthetic and Surgical Technique

Anesthetic management was performed using a standard protocol in our hospital. General anesthesia was induced with intravenous 100 μg of fentanyl and 1.2 mg/kg of propofol, followed by intravenous 1 mg/kg of rocuronium to facilitate endotracheal tube placement. General anesthesia was maintained with sevoflurane (2 to 3 volume%), nitrous oxide (1.8 L/min), and O2 (1.2 L/min). Intravenous 1 mg/kg rocuronium was administered, as required, to maintain adequate surgical relaxation. All patients underwent ultrasonography-guided right internal jugular vein catheterization after tracheal intubation in the operating room, and the position of the catheter was determined using a chest radiograph. Electrocardiogram, pulse oximetry, end-tidal carbon dioxide, invasive radial arterial pressure, CVP, and urine output were monitored. Fluid was not administered preoperatively and was restrictively infused after the start of anesthesia, maintaining CVP less than 5 mmHg until the hepatic parenchymal transection was complete. Thereafter, the crystalloid fluid was rapidly infused at 10 to 12 mL/kg/h to replace the surgical blood loss or fluid deficit, including insensible loss during the operation, and a colloid solution, hydroxyethyl starch (HES), was used considering volume status in the operating room. We used vasopressor drugs when the MAP decreased below 60 mmHg. Mostly 5 mg bolus of ephedrine was administered, but if an elevated heart rate was present, 50 mcg bolus of phenylephrine was injected. Red blood cells were transfused considering estimated blood loss and level of hemoglobin concentration (maintain ≥7 g/dL) in the operative room and if the hemoglobin concentration decreased to <7 g/dL in the postoperative period. ICU admission was decided if the patients required inotropic agents or had cardiac arrhythmia during the operation.

All hepatic resections were performed using the same hepatic parenchymal transection technique and the extent of hepatic resection was determined based on the tumor size and location. Parenchymal transection was performed using an ultrasonic aspirator, metal clips, and electrocautery device, and the cutting surface of the liver was sprayed with biological glue. Anatomic partial hepatectomy was performed in a standardized manner; however, if patients had poor liver function, non-anatomic partial hepatectomy was performed. All hepatic resections were performed by ligating the feeding vessels, and margins of at least 2 cm were secured.

### 2.4. Statistical Analysis

For intergroup comparisons, the data distribution was initially evaluated for normality with the Shapiro–Wilk test. Normally distributed data were presented as means ± standard deviations, and intergroup comparisons were conducted using the Student’s t-test or the Kruskal–Wallis test. Intergroup comparisons of descriptive data were conducted using the chi-square or Fisher’s exact test. The optimal cut-off value for the amount of IBL was determined by the area under the receiver operating characteristic analysis. Tumor recurrence and overall survival by operative factors were plotted using the Kaplan–Meier method and compared using the log-rank test. Multivariate analysis using the Cox proportional hazard regression method was performed to investigate the risk factors for tumor recurrence and overall survival. All statistical analyses were conducted using SPSS Statistics for Windows, version 19.0 (IBM Corp., Armonk, NY, USA).

## 3. Results

### 3.1. Clinico-Pathologic Findings of Patients

The clinic–pathologic characteristics of the participants are summarized in Table 1. There was no significant intergroup difference in age, sex, presence of HBV or HCV, BMI, and preoperative laboratory results, such as hemoglobin, total bilirubin, albumin, AFP, and PIVKA-II, except for the INR, which significantly increased in Group A compared to Group B (1.14 ± 0.11 vs. 1.09 ± 0.12, *p* = 0.031). In the pathologic findings, the tumor size (3.5 ± 2.2 vs. 3.1 ± 2.3, *p* = 0.321) was greater and microvascular invasion (20.0% vs. 14.0%, *p* = 0.365) was more common in Group A than in Group B; however, these differences were not significant. The presence of a poor histologic grade (III or IV) was significantly more common in Group A than in Group B (82.2% vs. 63.7%, *p* = 0.027).

### 3.2. Operative Outcomes of HCC Patients

Compared to Group B, the operative duration was significantly higher in Group A (244 ± 94 vs. 169 ± 58, *p* < 0.001), which may have been related to the higher IBL (1351 ± 698 vs. 354 ± 166, *p* < 0.001) and blood transfusion (63.8% vs. 6.3%, *p* < 0.001) in Group A. There was no significant intergroup difference in the proportion of major resection (68.1% vs. 68.4%, *p* = 0.968). The proportions of ICU admission (23.4% vs. 9.5%, *p* = 0.025) and hospital stay (15.1 ± 6.5 vs. 11.6 ± 4.7, *p* < 0.001) were significantly higher in Group A compared to Group B. There were no significant differences in the postoperative complications, including wound infection, pneumonia, acute kidney injury, liver insufficiency, abdominal wall hernia, and bile leakage (Table 2).

### 3.3. Cumulative Probability of Tumor Recurrence and Overall Survival by Operative Factors

The median follow-up period of the participants was 4.11 (0.54–11.2) years. The cumulative probability of tumor recurrence differed significantly (*p* = 0.001) according to the amount of IBL, as evidenced by the 1-, 3-, and 5-year recurrence-free survival rates for each group (Group A: 73.8%, 58.4%, and 46.1%, respectively; Group B: 92.5%, 78.2%, and 69.2%, respectively; Figure 1A). Patterns of post-resection tumor recurrence showed no significant intergroup differences in the number and site (Table 3). The overall survival rates at 1, 3, and 5 years were 100%, 97.0%, and 87.3% for Group A, and 100%, 100%, and 100% for Group B, respectively, with a significant intergroup difference (*p* = 0.017, Figure 1B). There was no significant difference in the cumulative probability of tumor recurrence (*p* = 0.181) and overall survival (*p* = 0.473) between the patients who did and did not receive intraoperative blood transfusion (Figure 2).

### 3.4. Risk Factor Analysis for Recurrence-Free Survival and Overall Survival after Resection

Significant risk factors for post-resection recurrence-free survival were albumin (*p* = 0.004), microvascular invasion (*p* = 0.008), and IBL ≥ 700 (*p* = 0.001) in the univariate analysis, and albumin (hazard ratio [HR], 0.471; 95% confidence interval [CI], 0.244–0.907, *p* = 0.024), microvascular invasion (HR, 2.616; 95% CI, 1.298–5.273, *p* = 0.007), and IBL ≥ 700 (HR, 2.325; 95% CI, 1.202–4.497, *p* = 0.012) in the multivariate analysis (Table 4).

Microvascular invasion was the only significant risk factor for post-resection overall survival in the multivariate analysis (HR, 4.695; 95% CI, 1.091–20.199, *p* = 0.038, Table 5).

## 4. Discussion

This study showed that not blood transfusion but excessive IBL was significantly associated with decreased disease-free and overall survival after surgical resection in HCC patients. Furthermore, in a multivariate analysis, excessive IBL was an independent prognostic factor for HCC recurrence after hepatic resection.

The adverse effect of an excessive IBL on the oncologic outcomes could be attributable to several possible reasons, as suggested with regard to other gastrointestinal solid tumors. First, large-volume IBL may impede the immune reaction against tumor cells, thereby increasing the likelihood of tumor recurrence. Excessive IBL increased the loss of plasma constituents such that the natural killer cell activity was decreased significantly in patients who underwent gastrointestinal surgery [16]. Moreover, the prevalence of tumor recurrence in the lung, a major site of hematogenous metastasis, was significantly higher in patients with gastric cancer who had excessive IBL, which supports the above-described association of IBL with oncologic outcomes [17,18]. Second, excessive IBL can induce hypoxic ischemia, which is accompanied by local and systemic inflammatory reactions that increase postoperative complications and induce greater antitumor immunosuppression [19]. In this study, the group of patients with IBL ≥ 700 mL had a significantly higher proportion of ICU admission and hospital stay than that of patients with IBL < 700 mL. Third, IBL may promote tumor-cell spillage into the peritoneal cavity, which might also contribute to tumor recurrence. Excessive IBL is an independent risk factor for intraperitoneal tumor recurrence in patients with gastric cancer [20,21]. However, only one patient in the group of patients with IBL ≥ 700 mL had intraperitoneal tumor recurrence, and this difference lacked statistical significance in this study (*p* = 0.927).

The influence of blood transfusion during HCC resection on long-term survival is still controversial. Several studies have shown that blood transfusion promotes HCC recurrence and decreases overall survival after hepatic resection and constitutes a significant risk factor for poor prognosis [11,12], possibly through the associated effects on immunosuppression [22]. A reduction in the activities of natural killer cells and helper T cells as well as an increase in suppressor T-cell activity and decrease in the absolute peripheral blood lymphocyte count were observed in patients who received blood transfusions [23,24]. Autologous rather than allogenic blood transfusion is recommended in HCC patients because of the minimal effect on cellular immunity and possible improvement of immunosuppression during the postoperative period [25]. In contrast, recent studies have found no association of blood transfusion with post-resection disease-free and overall survival rates in HCC [22,26]. Similarly, the present study showed that blood transfusion did not significantly influence the oncologic outcomes of patients who underwent hepatic resection.

Preoperative liver function could be closely related to increased IBL during hepatic resection. In this study, the baseline laboratory value for liver function was the INR, which increased significantly in patients with IBL ≥ 700 mL compared to those with IBL < 700 mL. However, there was no significant intergroup difference in the total bilirubin and albumin levels, which might be attributable to the eligibility criteria in this study, which excluded patients with borderline liver function. A previous study found that tumor size (>5 cm) and the type of hepatectomy (hemi-hepatectomy) necessitated a longer operative time and were significantly associated with IBL [13]. In this study, a significantly prolonged operative duration was observed in patients with excessive IBL; however, the tumor size and major resection did not have significant intergroup differences.

The strategy of maintaining low CVP during hepatic resection is widely employed to minimize IBL as it leads to less blood loss and easier control of bleeding from the hepatic venous injury during parenchymal transection [27]. This strategy seems to be especially important in obese patients who are more likely to experience excessive IBL during hepatic resection. Obese patients with low CVP showed significantly lower EBL than those with high CVP but a similar EBL to non-obese patients [28]. In this study, all patients received preoperative fluid restriction, which is one of the most effective and commonly used methods for decreasing the CVP. Various methods for decreasing CVP in patients with elevated CVP in the operating room have been reported previously; for example, changes in the patient’s body position: a head up tilt position significantly decreased CVP, but not hepatic vein pressure, which indicates that it may not be effective at reducing blood loss during hepatic resection [29]. The head up tilt position has also been reported to increase the risk of venous thromboembolic events, especially in patients with fluid restriction [25]. Diuretics and vasoactive drugs can also be administered to decrease CVP, but their effectiveness might be limited due to the time it takes for the onset of diuretics and decreased renal function during general anesthesia. A previous study showed that vasodilators, including nitroglycerin and esmolol, decreased systolic arterial pressure, but were not associated with blood loss from the hepatic surgical field because more than 60% of the blood supply is from the portal vein [30]. Milrinone, a cellular phosphodiesterase III inhibitor, has been suggested to decrease surgical field bleeding with a dynamic reduction in the CVP by a systemic vasodilation and enhancement of cardiac functions, including systolic contraction and diastolic relaxation [31]. Inferior vena cava clamping also reduces CVP, but it is not widely used because of the technical difficulties of dissecting the inferior vena cava and major hepatic veins, hemodynamic intolerance in some patients, and higher rates of thromboembolic events. We previously introduced bioelectrical impedance analysis for the assessment of preoperative volemia because of advantages such as noninvasiveness, rapid processing, ease of handling, and a relatively inexpensive cost [32]. It can also improve operative outcomes by creating a favorable surgical environment in living donor hepatectomy [33]. Hypovolemic phlebotomy has been proposed as an option for reducing the circulating blood volume; however, this method induces blood loss and may sometimes necessitate a blood transfusion that may also adversely affect the oncologic outcomes [34,35].

Despite excessive IBL being an independent prognostic factor for HCC recurrence after hepatic resection, it did not influence the overall survival. This might be due to the aggressive approach for the surveillance and treatment of recurrence. We generally perform CT or MRI and determine the levels of serum AFP and PIVKA-II every 3 months after surgical resection. That effective treatment, such as hepatic resection, RFA, or TACE, can be considered as early as possible before the progression of the recurrent tumor. Not all patients with tumor recurrence were effectively treated, and some cases of recurred tumor rapidly progressed despite the use of aggressive treatment.

There are some limitations of this study. The retrospective design of this study ensures that the accuracy of the data analyzed relies on the completeness of the medical records that were maintained at the hospital. The IBL might be inaccurate because it was calculated by the amount of fluid collected in a suction bottle, which contained not only blood but also other fluids, such as irrigating saline or peritoneal fluid, and the volume lost in surgical sponges was not taken into account. Furthermore, the possible reasons for tumor recurrence in patients with excessive IBL during hepatic resection will be analyzed in future studies to support the results of this study. Finally, the study population was relatively small. Therefore, further large-scale prospective studies are required to clarify the influence of the IBL and blood transfusion on the oncologic outcomes.

## 5. Conclusions

In summary, patients with IBL ≥ 700 mL had significantly higher blood transfusion requirements and a prolonged operative duration that proportionally increased the possibility of ICU admission and a longer hospital stay compared to those in patients with IBL < 700 mL. Excessive IBL, but not blood transfusion, was significantly associated with poor oncologic outcomes, including decreased disease-free and overall survival after surgical resection in HCC patients. In a multivariate analysis, IBL ≥ 700 mL, preoperative albumin level, and microvascular invasion were significant prognostic factors for HCC recurrence. Therefore, efforts to minimize IBL during hepatic resection are important for improving the long-term post-resection prognosis of HCC patients.

## Figures and Tables

**Figure 1 jpm-13-01115-f001:**
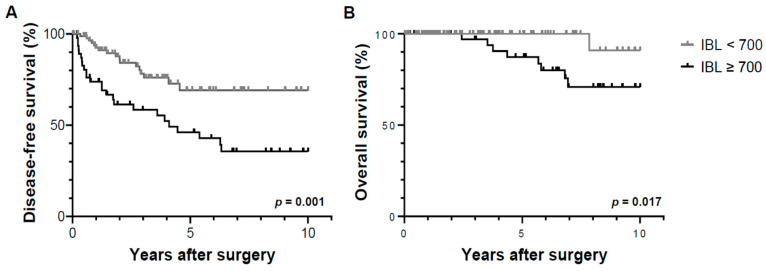
Disease-free survival (**A**) and overall survival (**B**) based on the amount of IBL. Compared to patients with IBL < 700 mL, the frequency of tumor recurrence was significantly higher in patients with IBL ≥ 700 mL (*p* = 0.001), with a significant intergroup difference in the overall survival (*p* = 0.017). IBL, intraoperative blood loss.

**Figure 2 jpm-13-01115-f002:**
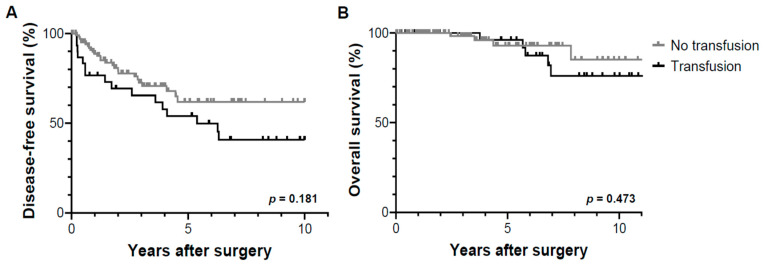
Disease-free survival (**A**) and overall survival (**B**) based on the need for blood transfusion. There was no significant intergroup difference in tumor recurrence (*p* = 0.181) and overall survival (*p* = 0.473) in patients with and without blood transfusion.

**Table 1 jpm-13-01115-t001:** Clinico-pathologic characteristics of the study cohort.

Variables	IBL ≥ 700 (n = 47)	IBL < 700 (n = 95)	*p* Value
Age, years	60.1 (±10.7)	62.2 (±10.8)	0.270
Male	40 (85.1%)	70 (73.7%)	0.125
Presence of HBV	27 (57.4%)	48 (50.5%)	0.437
Presence of HCV	6 (12.8%)	7 (7.4%)	0.294
BMI	25.6 (±4.5)	24.5 (±3.5)	0.113
Hemoglobin	13.5 (±1.7)	13.1 (±2.0)	0.226
Total bilirubin	0.8 (±0.3)	0.7 (±0.4)	0.058
Albumin	4.0 (±0.4)	4.2 (±0.4)	0.074
PT-INR	1.14 (±0.11)	1.09 (±0.12)	0.031
Tumor size	3.5 (±2.2)	3.1 (±2.3)	0.321
AFP	78 (±237)	125 (±349)	0.414
PIVKA-II	492 (±1612)	368 (±1140)	0.622
Microvascular invasion	9 (20.0%)	13 (14.0%)	0.365
Poor histologic grade (III or IV)	37 (82.2%)	58 (63.7%)	0.027

Data are presented as the mean ± standard deviations or numbers with percentages in parentheses unless otherwise indicated. Abbreviations: IBL, intraoperative blood loss; HBV, hepatitis B virus; HCV, hepatitis C virus; BMI, body mass index; PT-INR, prothrombin time–international normalized ratio; AFP, α-fetoprotein; PIVKA-II, prothrombin induced by vitamin K absence-II.

**Table 2 jpm-13-01115-t002:** Operative outcomes stratified by the study groups.

	IBL ≥ 700 (n = 47)	IBL < 700 (n = 95)	*p* Value
Operative duration	244 (±94)	169 (±58)	<0.001
Major resection	32 (68.1%)	65 (68.4%)	0.968
IBL	1351 (±698)	354 (±166)	<0.001
Blood transfusion	30 (63.8%)	6 (6.3%)	<0.001
ICU admission	11 (23.4%)	9 (9.5%)	0.025
Postoperative complications			
Wound infection	3 (6.4%)	8 (8.4%)	0.669
Pneumonia	1 (2.1%)	2 (2.1%)	0.993
Acute kidney injury	5 (10.6%)	7 (7.4%)	0.510
Liver insufficiency	3 (6.4%)	5 (5.3%)	0.785
Abdominal wall hernia	1 (2.1%)	1 (1.1%)	0.609
Bile leakage	1 (2.1%)	0	0.154
Hospital stay	15.1 (±6.5)	11.6 (±4.7)	<0.001

Data are presented as the mean ± standard deviations or numbers with percentages in parentheses unless otherwise indicated. Abbreviation: IBL, intraoperative blood loss; ICU, intensive care unit.

**Table 3 jpm-13-01115-t003:** Patterns of tumor recurrence after surgical resection in the study groups.

	IBL ≥ 700 (n = 47)	IBL < 700 (n = 95)	*p* Value
Recurrence	25 (53.2%)	17 (17.9%)	<0.001
Number			
Single	13 (52.0%)	8 (47.1%)	
Multiple	12 (48.0%)	9 (52.9%)	0.753
Site			
Intrahepatic	20 (80.0%)	13 (76.5%)	
Extrahepatic	5 (20.0%)	4 (23.5%)	0.784

**Table 4 jpm-13-01115-t004:** Risk factor analysis for tumor recurrence after surgical resection.

Variable	Univariate Analysis		Multivariate Analysis	
	Hazard Ratio (95% CI)	*p* Value	Hazard Ratio (95% CI)	*p* Value
Age ≥ 60 years	0.621 (0.333–1.159)	0.134		
Male	0.529 (0.223–1.258)	0.150		
Presence of HBV	0.720 (0.392–1.321)	0.289		
Presence of HCV	2.519 (1.111–5.709)	0.027		
BMI ≥ 25	1.170 (0.637–2.150)	0.613		
Total bilirubin	1.463 (0.704–3.038)	0.308		
Albumin	0.402 (0.217–0.742)	0.004	0.471 (0.244–0.907)	0.024
PT-INR	4.707 (0.420–52.783)	0.209		
Tumor size ≥ 5 cm	0.499 (0.154–1.616)	0.246		
AFP ≥ 400	1.347 (0.480–3.783)	0.572		
PIVKA-II ≥ 80	0.828 (0.412–1.666)	0.597		
Microvascular invasion	2.594 (1.288–5.222)	0.008	2.616 (1.298–5.273)	0.007
Poor histologic grade (III or IV)	1.426 (0.694–2.931)	0.334		
Major resection	0.676 (0.332–1.379)	0.282		
IBL ≥ 700 mL	2.788 (1.500–5.181)	0.001	2.325 (1.202–4.497)	0.012
Transfusion	1.528 (0.818–2.857)	0.184		

Data are presented as the mean ± standard deviations or numbers with percentages in parentheses unless otherwise indicated. Abbreviation: HBV, hepatitis B virus; HCV, hepatitis C virus; BMI, body mass index; PT-INR, prothrombin time–international normalized ratio; AFP, α-fetoprotein; PIVKA-II, prothrombin induced by vitamin K absence-II; IBL, intraoperative blood loss; CI, confidence interval.

**Table 5 jpm-13-01115-t005:** Risk factor analysis for overall survival after surgical resection.

Variable	Univariate Analysis		Multivariate Analysis	
	Hazard Ratio (95% CI)	*p* Value	Hazard Ratio (95% CI)	*p* Value
Age ≥ 60 years	1.188 (0.318–4.434)	0.798		
Male	0.627 (0.077–5.106)	0.663		
Presence of HBV	0.365 (0.091–1.461)	0.154		
Presence of HCV	2.203 (0.453–10.713)	0.328		
BMI ≥ 25	2.630 (0.654–10.568)	0.173		
Total bilirubin	1.084 (0.256–4.597)	0.912		
Albumin	0.509 (0.139–1.865)	0.308		
PT-INR	10.127 (0.126–814.239)	0.301		
Tumor size ≥ 5 cm	0.043 (0.000–1729.022)	0.561		
AFP ≥ 400	1.772 (0.218–14.392)	0.593		
PIVKA-II ≥ 80	0.580 (0.117–2.885)	0.506		
Microvascular invasion	4.214 (0.992–17.897)	0.051	4.695 (1.091–20.199)	0.038
Poor histologic grade (III or IV)	3.272 (0.402–26.609)	0.268		
Major resection	0.353 (0.044–2.833)	0.327		
IBL ≥ 700 mL	8.390 (1.044–67.408)	0.045		
Transfusion	1.618 (0.430–6.090)	0.477		

Data are presented as the mean ± standard deviations or numbers with percentages in parentheses unless otherwise indicated. Abbreviation: HBV, hepatitis B virus; HCV, hepatitis C virus; BMI, Body mass index; PT-INR, Prothrombin time–international normalized ratio; AFP, α-fetoprotein; PIVKA-II, prothrombin induced by vitamin K absence-II; IBL, intraoperative blood loss; CI, confidence interval.

## Data Availability

Data available from the authors upon request.

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
