# Peer review of "Influence of Intraoperative Blood Loss on Tumor Recurrence after Surgical Resection in Hepatocellular Carcinoma"

_jpm, 2023, doi:10.3390/jpm13071115_

Round 1
Reviewer 1 Report
Well-conducted study on the impact of intraoperative blood loss during hepatic resections on tumor recurrence and survival. Useful information will help to use measures to reduce operative blood loss and improve survival.
Author Response
Well-conducted study on the impact of intraoperative blood loss during hepatic resections on tumor recurrence and survival. Useful information will help to use measures to reduce operative blood loss and improve survival.
Ans) I’m thank the reviewer for going through our manuscript carefully. I appreciate your good comments.

Reviewer 2 Report
This interesting study highlights intra-operative blood loss (IBL) as a predictor of tumor recurrence.
There are several critical fallacies in the study, such as:
1) Measurement of the main comparator variable IBL with a cut-off of 700 mL seems inaccurate as only the suction volume has been taken into account. As correctly mentioned in the paper, this overestimates blood loss as the volume of irrigation has not been subtracted from this. On the other hand, an estimate of the volume lost in surgical sponges has not been taken into account, which is common practice. Therefore, unless more accurate estimates of IBL are used, the conclusions may not be valid.
2) The rationale for using 700 mL as the cut-off for IBL is not elaborated.
3) The threshold for transfusion was hemoglobin of 7 g/dl, but the hemoglobin before the surgery has not been mentioned.
4) The study is unable to support the hypothesis that IBL causes higher recurrence due to lower immune reaction, especially without the need for more blood transfusions. Since there was only one patient with intra-peritoneal recurrence, it does not support the tumor spillage into the peritoneal cavity hypothesis. Even the hypoxic ischemia being the cause of higher recurrence is unknown as patients with higher IBL may not have been hypoxic or hypotensive. It would be unfair to base the most important conclusion of the study on these assumptions. This is especially important because the diference was observed only with IBL and not with transfusions.
5) Since the group with higher recurrence also had a significantly higher proportion of poor histological-grade tumors, the groups were not comparable and therefore the conclusions of this study may be due to its confounding effect.
6) Microvascular invasion is a well-known risk factor for tumor recurrence, which was also found in this study.
7) It is also possible that tumors with higher IBL may have been larger, closer to vital structures requiring larger resections, some of which could also be the reason for higher recurrence. This important information is missing from the manuscript.
8) Although it is mentioned that a 2cm margin was achieved in all cases, being a retrospective study, histopathology reports could be evaluated to look for any cases where R0 resection was not achieved, which would be an important reason for recurrence.
Besides tumor-related factors, operative factors, as perioperative blood transfusion has been reported to be related with HCC recurrence could be written as Besides tumor-related factors, operative factors, such as perioperative blood transfusion has been reported to be related to HCC recurrence
However, excessive intra-operative blood loss (IBL) always predisposes to blood transfusion that IBL itself may also influence on oncologic outcomes do the authors want to say IBL and blood transfusion may influence oncologic outcomes
The clinic-pathologic findings, operative outcomes, and the cumulative probability of tumor recurrence and overall survival were compared between two groups could be written as The clinico-pathologic findings, operative outcomes, and the cumulative probability of tumor recurrence and overall survival were compared between the two groups
Surgical resection is the only curative treatment among various loco-regional therapies for HCC that has shown favorable survival could be written as Surgical resection is the only curative treatment among various loco-regional therapies for HCC that has have shown favorable survival
Clinicopathologic factors related to tumor could be written as Clinicopathologic factors related to the tumor
Despite the recent improvements in surgical techniques, perioperative management, and dissection devices for hepatic resection, the intraoperative blood loss could be written as Despite the recent improvements in surgical techniques, perioperative management, and dissection devices for hepatic resection, the intra-operative blood loss
long term could be written as long-term
We used vasopressor drug could be written as We used vasopressor drugs
50 mcg bolus of phenylephrine were injected could be written as 50 mcg bolus of phenylephrine was injected
Anatomic partial hepatectomy was performed in a standardized manner; however, if the patients had poor liver function, non-anatomic partial hepatectomy was also performed could be written as Anatomic partial hepatectomy was performed in a standardized manner; however, if the patients had poor liver function, non-anatomic partial hepatectomy was also performed
clinico-pathological is written as clinic-pathological at a few places
Author Response
This interesting study highlights intra-operative blood loss (IBL) as a predictor of tumor recurrence.
Ans) I’m thank the reviewer for going through our manuscript carefully and suggesting points to improve the same.
There are several critical fallacies in the study, such as:
1) Measurement of the main comparator variable IBL with a cut-off of 700 mL seems inaccurate as only the suction volume has been taken into account. As correctly mentioned in the paper, this overestimates blood loss as the volume of irrigation has not been subtracted from this. On the other hand, an estimate of the volume lost in surgical sponges has not been taken into account, which is common practice. Therefore, unless more accurate estimates of IBL are used, the conclusions may not be valid.
Ans) I also agree with your opinion. IBL might be inaccurate because in a suction bottle, which contained not only blood but also other fluids, such as irrigating saline or peritoneal fluid, and the volume lost in surgical sponges has not been taken into account. A 700 mL might be an inaccurate amount of blood, but it can show the degree of bleeding as small or large during hepatic resection that we compared using this value. This study was a retrospective nature and it might be a limitation of this study that we added this limitation in the discussion section as below.
‘IBL might be inaccurate because it was calculated by the amount of fluid collected in a suction bottle, which contained not only blood but also other fluids, such as irrigating saline or peritoneal fluid, and the volume lost in surgical sponges has not been taken into account.’ (294-297)
2) The rationale for using 700 mL as the cut-off for IBL is not elaborated.
Ans) A 700 mL might be an inaccurate amount of blood, but it can show the degree of bleeding as small or large during hepatic resection that we compared using this value. The optimal cut-off value
for the amount of IBL was determined by the area under the receiver operating characteristic analysis. We added the comment in the method section.
‘The optimal cut-off value for the amount of IBL was determined by the area under the receiver operating characteristic analysis.’ (129-130)
3) The threshold for transfusion was hemoglobin of 7 g/dl, but the hemoglobin before the surgery has not been mentioned.
Ans) There was no significant intergroup difference in preoperative hemoglobin (13.5 ±1.7 vs. 13.1 ±2.0, p =0.226). We added the level of hemoglobin in the result section and Table 1.
4) The study is unable to support the hypothesis that IBL causes higher recurrence due to lower immune reaction, especially without the need for more blood transfusions. Since there was only one patient with intra-peritoneal recurrence, it does not support the tumor spillage into the peritoneal cavity hypothesis. Even the hypoxic ischemia being the cause of higher recurrence is unknown as patients with higher IBL may not have been hypoxic or hypotensive. It would be unfair to base the most important conclusion of the study on these assumptions. This is especially important because the difference was observed only with IBL and not with transfusions.
Ans) Blood transfusion is previously reported to be a significant risk factor for tumor recurrence. However, there has been little study which focused on the IBL itself. We performed this study because blood transfusion is always required for excessive IBL and, therefore, the associated factors may be closely related, with considerable overlap of the adverse effects. These reasons are the possible reasons, and further studies might be required to reveal the cause of tumor recurrence in case of excessive bleeding during hepatic resection. We added this limitation in the discussion section as below.
‘Furthermore, the possible reasons for tumor recurrence in patients with excessive IBL during hepatic resection would be analyzed in future studies to support the results of this study.’ (297-299)
5) Since the group with higher recurrence also had a significantly higher proportion of poor histological-grade tumors, the groups were not comparable and therefore the conclusions of this study may be due to its confounding effect.
Ans) We investigated the effect of IBL on tumor recurrence after hepatic resection using a multivariate analysis to minimize confounding effect. Table 4 showed that not poor histologic grade, but albumin, microvascular invasion and IBL ≥700 mL were the significant risk factors for tumor recurrence.
6) Microvascular invasion is a well-known risk factor for tumor recurrence, which was also found in this study.
Ans) Table 4 showed that microvascular invasion and IBL ≥700 mL were the significant risk factors for tumor recurrence in multivariate analysis.
7) It is also possible that tumors with higher IBL may have been larger, closer to vital structures requiring larger resections, some of which could also be the reason for higher recurrence. This important information is missing from the manuscript.
Ans) I agree with your opinion. Table 1 showed that there was no significant difference of the proportion of major resection between the groups (68.1% vs. 68.4%, p= 0.968).
8) Although it is mentioned that a 2cm margin was achieved in all cases, being a retrospective study, histopathology reports could be evaluated to look for any cases where R0 resection was not achieved, which would be an important reason for recurrence.
Ans) Surgical margin would be the most important factor for tumor recurrence. We only analyzed patients who underwent curative surgical resection of newly diagnosed single HCC. We added this comment in the method section.
‘We enrolled 142 participants with well-preserved liver function who underwent cu-rative surgical resection of newly diagnosed single HCC between March 2010 and July 2021 at our hospital.’ (56-57)
Comments on the Quality of English Language
Besides tumor-related factors, operative factors, as perioperative blood transfusion has been reported to be related with HCC recurrence could be written as Besides tumor-related factors, operative factors, such as perioperative blood transfusion has been reported to be related to HCC recurrence
However, excessive intra-operative blood loss (IBL) always predisposes to blood transfusion that IBL itself may also influence on oncologic outcomes do the authors want to say IBL and blood transfusion may influence oncologic outcomes
The clinic-pathologic findings, operative outcomes, and the cumulative probability of tumor recurrence and overall survival were compared between two groups could be written as The clinico-pathologic findings, operative outcomes, and the cumulative probability of tumor recurrence and overall survival were compared between the two groups
Surgical resection is the only curative treatment among various loco-regional therapies for HCC that has shown favorable survival could be written as Surgical resection is the only curative treatment among various loco-regional therapies for HCC that has have shown favorable survival
Clinicopathologic factors related to tumor could be written as Clinicopathologic factors related to the tumor
Despite the recent improvements in surgical techniques, perioperative management, and dissection devices for hepatic resection, the intraoperative blood loss could be written as Despite the recent improvements in surgical techniques, perioperative management, and dissection devices for hepatic resection, the intra-operative blood loss
long term could be written as long-term
We used vasopressor drug could be written as We used vasopressor drugs
50 mcg bolus of phenylephrine were injected could be written as 50 mcg bolus of phenylephrine was injected
Anatomic partial hepatectomy was performed in a standardized manner; however, if the patients had poor liver function, non-anatomic partial hepatectomy was also performed could be written as Anatomic partial hepatectomy was performed in a standardized manner; however, if the patients had poor liver function, non-anatomic partial hepatectomy was also performed
clinico-pathological is written as clinic-pathological at a few places
Ans) We corrected the words in the manuscript following your opinion

Reviewer 3 Report
The authors investigated the influence of operative factors including intraoperative blood loss and perioperative blood transfusion on long term oncologic outcomes, including tumor recurrence and overall survival, after surgical resection in hepatocellular carcinoma patients. Authors have done good job in formulating the idea and summarizing their findings. Study results validate authors discussion and conclusions. However, I have few suggestions.
1. Include primary and secondary study outcomes in detail in the Methods section.
2. Explain why a IBL cut off of 700 mL was chosen.
3. Study major perioperative complications such as bile leakage, ascites, acute renal failure, etc between the groups and see if that translates into prolonged hospital stay and mortality in the respective groups.
4. Excessive IBL was an independent prognostic factor for HCC recurrence after hepatic resection but not for overall survival after surgical resection. Explain why increased recurrence did not translate into worse survival.
5. Several other variables could have influenced the primary outcomes in addition to excessive IBL. Consider doing a propensity score matching analysis model and adjusted logistic regression to avoid study bias.
Author Response
The authors investigated the influence of operative factors including intraoperative blood loss and perioperative blood transfusion on long term oncologic outcomes, including tumor recurrence and overall survival, after surgical resection in hepatocellular carcinoma patients. Authors have done good job in formulating the idea and summarizing their findings. Study results validate authors discussion and conclusions. However, I have few suggestions.
Ans) I’m thank the reviewer for going through our manuscript carefully and suggesting points to improve the same.
1. Include primary and secondary study outcomes in detail in the Methods section.
Ans) We added the primary and secondary outcomes of this study in the method section as below.
‘The primary outcomes outcome of this study was to compare the cumulative probability of tumor recurrence and overall survival between the two groups. And the secondary out-comes were to identify risk factors after adjusting for several parameters that could poten-tially influence tumor recurrence and overall survival after surgical resection for HCC.’ (61-66)
2. Explain why a IBL cut off of 700 mL was chosen.
Ans) The optimal cut-off value for the amount of IBL was determined by the area under the receiver operating characteristic analysis. We added the comment in the method section.
‘The optimal cut-off value for the amount of IBL was determined by the area under the receiver operating characteristic analysis.’ (129-130)
3. Study major perioperative complications such as bile leakage, ascites, acute renal failure, etc between the groups and see if that translates into prolonged hospital stay and mortality in the respective groups.
Ans) We re-analyzed the postoperative outcomes between the two groups. However there were no significant differences in the postoperative complications including wound infection, pneumonia, acute kidney injury, liver insufficiency, abdominal wall hernia, and bile leakage. We added the definition of acute kidney injury and liver insufficiency in the method section and the postoperative complicaitons in the result section as below.
‘Acute kidney injury (AKI) was defined in accordance with the 2012 Kidney Disease Im-proving Global Outcomes guidelines [14] which had higher predictability than other crite-ria for assessing prognosis: increase in serum creatinine by ≥0.3 mg/dl within 48 hours; increase in serum creatinine to ≥1.5 times baseline within 7 days before surgery; or urine volume <0.5 mL/kg/h for 6 hours. Postoperative liver insufficiency was defined as a peak postoperative TB level of >7 mg/dL and/or the presence of ascites >500 mL/day based on a previous study [15].’ (81-87)
References
14. Khwaja, A. KDIGO clinical practice guidelines for acute kidney injury. Nephron Clin Pract. 2012, 120, c179-84.
15. Mullen, J.T.; Ribero, D.; Reddy, S.K.; Donadon, M.; Zorzi, D.; Gautam, S.; Abdalla, E.K.; Curley, S.A.; Capussotti, L.; Clary, B.M.; et al. Hepatic insufficiency and mortality in 1,059 noncirrhotic patients undergoing major hepatectomy. J Am Coll Surg. 2007, 204, 854-64.
‘There were no significant differences in the postoperative complications including wound infection, pneumonia, acute kidney injury, liver insufficiency, abdominal wall hernia, and bile leakage (Table 2).’ (159-161)
4. Excessive IBL was an independent prognostic factor for HCC recurrence after hepatic resection but not for overall survival after surgical resection. Explain why increased recurrence did not translate into worse survival.
Ans) Despite excessive IBL was an independent prognostic factor for HCC recurrence after hepatic resection, it was not influenced on overall survival in this study. This might be due to the aggressive approach for the surveillance and treatment of recurrence. We generally perform CT or MRI and determine the levels of serum AFP and PIVKA-II every 3 months after surgical resection that effective treatment, such as hepatic resection, RFA, or TACE, can be considered as early as possible before progression of the recurrent tumor. Not all patients with tumor recurrence were effectively treated, and some cases of recurred tumor rapidly progressed despite the use of aggressive treatment. We added these comments in the discussion section.
‘Despite excessive IBL was an independent prognostic factor for HCC recurrence after hepatic resection, it was not influenced on overall survival in this study. This might be due to the aggressive approach for the surveillance and treatment of recurrence. We generally perform CT or MRI and
determine the levels of serum AFP and PIVKA-II every 3 months after surgical resection that effective treatment, such as hepatic resection, RFA, or TACE, can be considered as early as possible before progression of the recurrent tumor. Not all patients with tumor recurrence were effectively treated, and some cases of recurred tumor rapidly progressed despite the use of aggressive treatment.’ (286-293)
5. Several other variables could have influenced the primary outcomes in addition to excessive IBL. Consider doing a propensity score matching analysis model and adjusted logistic regression to avoid study bias.
Ans) We already showed the results of multivariate analysis using the Cox proportional hazard regression method to adjust study bias in Table 4 and Table 5. Significant risk factors for post-resection recurrence-free survival were albumin (p =0.004), microvascular invasion (p =0.008), and IBL ≥700 (p =0.001), in the univariate analysis, and albumin (hazard ratio [HR], 0.471; 95% confidence interval [CI], 0.244–0.907, p =0.024), microvascular invasion (HR, 2.616; 95% CI, 1.298–5.273, p =0.007), and IBL ≥700 (HR, 2.325; 95% CI, 1.202–4.497, p =0.012), in the multivariate analysis (Table 4). And microvascular invasion was the only significant risk factor for post-resection overall survival in the multivariate analysis (HR, 4.695; 95% CI, 1.091–20.199, p =0.038, Table 5).

Round 2
Reviewer 2 Report
Th
Thanks for making the changes
Reviewer 3 Report
Authors addressed my questions. This manuscript can be published if other reviewers questions are addressed since they raised very important concerns with the study.